# Chronic Kidney Disease-Induced Arterial Media Calcification in Rats Prevented by Tissue Non-Specific Alkaline Phosphatase Substrate Supplementation Rather Than Inhibition of the Enzyme

**DOI:** 10.3390/pharmaceutics13081138

**Published:** 2021-07-26

**Authors:** Britt Opdebeeck, Ellen Neven, José Luis Millán, Anthony B. Pinkerton, Patrick C. D’Haese, Anja Verhulst

**Affiliations:** 1Laboratory of Pathophysiology, Department of Biomedical Sciences, University of Antwerp, 2610 Wilrijk, Belgium; britt.opdebeeck2@uantwerpen.be (B.O.); ellen.neven@uantwerpen.be (E.N.); anja.verhulst@uantwerpen.be (A.V.); 2Sanford Burnham Prebys Medical Discovery Institute, La Jolla, CA 92037, USA; millan@sbpdiscovery.org (J.L.M.); apinkerton@sbpdiscovery.org (A.B.P.)

**Keywords:** arterial calcification, mineral bone disorder, alkaline phosphatase, chronic kidney disease, pyrophosphate

## Abstract

Patients with chronic kidney disease (CKD) suffer from arterial media calcification and a disturbed bone metabolism. Tissue-nonspecific alkaline phosphatase (TNAP) hydrolyzes the calcification inhibitor pyrophosphate (PPi) into inorganic phosphate (Pi) and thereby stimulates arterial media calcification as well as physiological bone mineralization. This study investigates whether the TNAP inhibitor SBI-425, PPi or the combination of both inhibit arterial media calcification in an 0.75% adenine rat model of CKD. Treatments started with the induction of CKD, including (i) rats with normal renal function (control diet) treated with vehicle and CKD rats treated with either (ii) vehicle, (iii) 10 mg/kg/day SBI-425, (iv) 120 µmol/kg/day PPi and (v) 120 µmol/kg/day PPi and 10 mg/kg/day SBI-425. All CKD groups developed a stable chronic renal failure reflected by hyperphosphatemia, hypocalcemia and high serum creatinine levels. CKD induced arterial media calcification and bone metabolic defects. All treatments, except for SBI-425 alone, blocked CKD-related arterial media calcification. More important, SBI-425 alone and in combination with PPi increased osteoid area pointing to a less efficient bone mineralization. Clearly, potential side effects on bone mineralization will need to be assessed in any clinical trial aimed at modifying the Pi/PPi ratio in CKD patients who already suffer from a compromised bone status.

## 1. Introduction

Chronic kidney disease (CKD) is called a “disease multiplier” as multiple comorbidities occur in these patients including a high risk for cardiovascular disease. Arterial media calcification or the deposition of calcium-phosphate crystals (i.e., hydroxyapatite) in the medial layer of the arterial wall is a prevalent cardiovascular complication in CKD patients. Calcification of the arterial wall is a life-threating condition as arterial compliance decreases and arterial stiffness increases, which is followed by an inadequate peripheral blood flow, ultimately leading to complications including left ventricular hypertrophy, sudden cardiac arrest and heart failure [1]. Therapies targeting arterial media calcification are sparse and focus mainly on maintaining a good balance in mineral homeostasis, which may be achieved by, e.g., the use of phosphate binders [2]. However, the pathology of arterial media calcification is multifactorial. Besides a passive component, spontaneous calcium-phosphate precipitation in the arteries due to high levels of calcium and phosphate ions in the blood compartment, an active cell-regulated component also occurs. Vascular smooth muscle cells (VSMCs), the major cell type in the medial layer of the vessel wall, retain the capacity to transdifferentiate into osteo/chondrogenic-like cells with the upregulation of bone-like genes including Runt-related transcription factor 2 (*Runx2*), SRY-box transcription factor 9 (*Sox9*) and bone morphogenic protein 2 (*Bmp2*). Ultimately, transdifferentiated VSMCs secrete calcified, small membrane-enclosed extracellular vesicles or exosomes to provide mineral nucleation sites for the hydroxyapatite crystal growth [3,4]. Additionally, VSMCs are capable of sensing a degree of matrix stiffness, thereby inducing upregulation of the osteoblast specific proteins *Bmp2* and *Runx2* [5]. Increased arterial stiffness is a typical vasculopathy in CKD patients [6]. Besides this, CKD patients suffer from endothelial dysfunction (i.e., impaired nitric oxide regulation) favoring the calcification process in the arteries [7,8]. Furthermore, calcification inhibitors such as fetuin A, pyrophosphate (PPi) and Matrix gla protein (MGP) are produced in an attempt to restrict excessive calcification or mineralization. However, in the presence of CKD, these calcification inhibitors are decreased, leading to ectopic calcification or mineralization of the arterial wall [9,10].

Extracellular nucleotides (i.e., ATP and UTP) also influence the arterial calcification process through the interaction with ecto-nucleotidases such as ectonucleotide pyrophosphate/phosphodiesterase (NPP1), ectonucleoside triphosphate diphosphohydrolase 1 (NTPD1) and tissue-nonspecific alkaline phosphatase (TNAP). These metalloenzymes maintain the pyrophosphate (PPi)/inorganic phosphate (Pi) homeostasis through (i) NPP1-mediated hydrolysis of ATP into AMP + PPi, (ii) NTPD1-mediated hydrolysis of ATP/ADP into ADP/AMP + Pi and (iii) TNAP-mediated breakdown of PPi into Pi [11]. This PPi/Pi homeostasis plays a crucial role in the arterial media calcification process. A high serum Pi level is known as one of the most powerful inducers of this process, while PPi is a potent calcification inhibitor as it prevents the incorporation of Pi into the hydroxyapatite crystals and, by these means, halts their growth [12]. This is shown with genetic inactivation of PPi generation, which leads to genetic diseases with the development of arterial calcification (i.e., generalized arterial calcification of infancy) [13] but also in dialysis patients wherein low levels of PPi have been reported and are associated with the development of arterial media calcification [14,15]. The group of Millan et al. has developed a potent and specific TNAP inhibitor compound SBI-425 that increases PPi levels [16]. Studies in mice have shown that SBI-425 prevents the development of arterial calcification [17,18,19,20,21]. Furthermore, a recent study conducted by our group revealed, for the first time, that SBI-425 inhibited (warfarin-induced) arterial media calcification in the rat, which was, however, accompanied by mild inhibiting effects on bone formation and mineral apposition rates [22]. The latter can be attributed to the fact that arterial calcification resembles physiological bone mineralization, in which TNAP (by hydrolyzing PPi into Pi) also plays an important role [23], and should be taken into consideration when administered to CKD patients. CKD patients are, indeed, an important target population to be treated with SBI-425; they do, however, already suffer from bone diseases [24]. Based on this, the present study investigates the effects of SBI-425 on both arterial media calcification and bone metabolism in rats with adenine-induced CKD. Because the aortic calcifications in an adenine-induced CKD rat model develop faster and are more severe (four times higher calcification scores) as compared to the rat model with warfarin-induced arterial calcification [22] and normal renal function, combination therapy of SBI-425 with PPi supplementation was included in the study with the intention of increasing therapeutic efficacy.

## 2. Materials and Methods

### 2.1. Animal Experiment

All animal experiments were performed in accordance with the National Institutes of Health Guide for the Care and Use of Laboratory Animals 85–23 (1996) and approved by the University of Antwerp Ethics Committee (Permit number: 2017-05, Approval date: 23 February 2017). Animal experiments have been performed according to ARRIVE guidelines. Animals were housed two per cage, exposed to 12-h light/dark cycles, and had free access to food and water. A total of 52 male Wistar rats (225–250 g, Iffa Credo, Belgium) were randomly assigned to 5 study groups: (i) rats with normal renal function (control diet) treated with vehicle (*n* = 4), (ii) CKD rats treated with vehicle (*n* = 12), (iii) CKD rats treated with 10 mg/kg/day TNAP-inhibitor (*n* = 12) via intraperitoneal catheter (rounded polyurethane catheter ROPAC-3.5PR, Access technologies, Skokie, IL, USA), (iv) CKD rats treated with 120 µmol/kg/day PPi (*n* = 12) via intraperitoneal catheter and (v) CKD rats treated with 120 µmol/kg/day PPi and 10 mg/kg/day TNAP-inhibitor (*n* = 12) via intraperitoneal catheter. CKD was induced by administration of a high phosphate diet (SSNIFF, Spezialdiäten, Soest, Germany) for two weeks followed by a 0.75% adenine diet (SSNIFF, Spezialdiäten, Soest, Germany) for four weeks, after which animals were sacrificed. SBI-425, PPi and combined SBI-425/PPi treatments started concomitantly with the start of the dietary administration of 0.75% adenine diet until sacrifice (see study design in Appendix A). SBI-425 was synthesized at the Prebys Center for Drug Discovery, Sanford Burnham Prebys Medical Discovery Institute (La Jolla, CA, USA) via the published procedure [16]. One week before the start of the study, an i.p. catheter was placed into the peritoneal cavity of the rats. This procedure was performed as described in [22]. Dextrose (5% in PBS)/1% ethanol/0.3% sodium hydroxide solution was used as solvent for SBI-425 and PPi. The solvent without addition of SBI-425 or PPi comprised the vehicle treatment. Blood and urine samples were taken before the start, after two weeks of 0.75% adenine diet supplementation and at sacrifice (week 4). Hereto, metabolic cages were used to obtain 24 h urine samples. Subsequently, blood sampling via the tail vein of restrained, conscious animals was performed. Furthermore, 7 and 3 days before sacrifice, animals received an intravenous injection of 30 mg/kg tetracycline and 25 mg/kg demeclocycline to evaluate dynamic bone parameters. At the end of study, rats were sacrificed through exsanguination through the retro-orbital plexus after anesthesia with 80 mg/kg ketamine (Pfizer, Puurs, Belgium) and 10 mg/kg xylazine (Bayer Animal Health, Monheim, Germany) via intraperitoneal injection. Before the planned euthanasia, one vehicle-treated CKD rat died due to CKD.

### 2.2. Analysis of Biochemical Parameters

Phosphorus and calcium levels in serum and urine samples were obtained with, respectively, the Ecoline S Phosphate kit (Diasys, Holzheim, Germany) and flame atomic absorption spectrometry (FAAS) (Perkin-Elmer, Wellesley, MA, USA) after dilution in 0.1% La(NO_3_)_3_ to eliminate chemical interference. Serum alkaline phosphatase activity, alanine transaminase (ALT) and aspartate transaminase (AST) levels were measured by an auto-analyzer (Dimension Vista 1500 System, Siemens AG, München, Germany) at the Antwerp University Hospital, Wilrijk, Belgium. The Bowers and McComb [25] procedure was used to determine alkaline phosphatase activity. Briefly, substrate nitrophenyl phosphate was converted by alkaline phosphatase to a yellow chromogen *p*-nitrophenyl at pH 10.25. To determine aortic alkaline phosphatase activity, aortic tissue was homogenized in RIPA buffer (Sigma-Aldrich, St. Louis, MO, USA) supplemented with 2% TritonX (Sigma-Aldrich, St. Louis, MO, USA), 0.51 mg/mL aprotinin (MP Biomedicals, Solon, OH, USA) and 3.14 mg/mL benzamidine (Sigma-Aldrich, St. Louis, MO, USA) on ice. Subsequently, protein content and alkaline phosphatase activity of aortic lysates were analyzed through, respectively, the BCA assay (Thermo Fisher Scientific Inc., Waltham, MA, USA) and an auto-analyzer. PPi levels were analysed in plasma using a PPiLight inorganic pyrophosphate assay (Lonza, Basel, Switzerland). Purification of plasma samples was performed using centrifugal filters of 0.1 µM (Merck Millipore Ltd., County Cork, Ireland). Serum fibroblast growth factor 23 (FGF23) levels were measured by means of an ELISA (Kainos Laboratories, Tokyo, Japan).

### 2.3. Quantification of Arterial Calcification

Arterial calcifications were quantified via determining (i) the calcium content in the abdominal aorta and smaller arteries by use of FAAS method and (ii) percentage calcified area on Von Kossa stained sections of the thoracic aorta. First, the abdominal aorta, carotid and femoral arteries were weighed on a precision balance where after the samples were digested in 65% HNO_3_ at 60 °C for 6 h. After appropriate dilution in 0.1% La(NO_3_)_3_ to eliminate chemical interference, the calcium content in the digests was measured with FAAS and expressed as mg calcium/g wet tissue. Second, the thoracic aorta was fixed in neutral buffered formalin for 90 min. Approximately, 15–20 tissue parts of 2 to 3 mm were cut and embedded upright in a paraffin block. Aortic sections (4 µM thick) were stained with Von Kossa and counterstained with hematoxylin and eosin. Axiovision image analysis software (Release 4.5; Carl Zeiss, Oberkochen, Germany) was used to calculate the percentage calcified area, which equals the ratio of pixels of Von Kossa positive area versus the pixels of total tissue area.

### 2.4. Quantitative Real Time PCR

The mRNA transcript expression of glyceraldehyde 3-phosphate dehydrogenase (*Gapdh*) (Rn99999916_s1, Thermofisher), *Tnap* (Rn01516028_m1), smooth muscle 22-alpha (*Sm22-alpha*) (Rn01642285_g1), R*unx2* (Rn01512298_m1), ectonucleoside triphosphate diphosphohydrolase 1 (*Ntpd1*) (Rn00574887_m1), ectonucleotide pyrophosphatase/phosphodiesterase 1 (*Npp1*) (Rn01638706_m1) and 3 (*Npp3*) (Rn00571329_m1), phosphoethanolamine/phosphocholine phosphatase 1 (*Phospho1*) (Rn01496968_m1), SRY-box transcription factor 9 (*Sox9*) (Rn01751070_m1), sodium-dependent phosphate transporter 1 (*Pit-1*) (Rn00579811_m1) and *Pit-2* (Rn00568130_m1) was determined in the distal part of the abdominal aorta. Total mRNA was extracted using the RNeasy Fibrous Tissue mini kit (Qiagen, Hilden, Germany) and reverse transcribed to cDNA using the High-Capacity cDNA Archive kit (Applied Biosystems, Foster City, CA, USA). Real-time polymerase chain reaction (PCR) with a QuantStudio 3 Detection System (Thermo Fisher Scientific Inc., Waltham, MA, USA) based on TaqMan fluorescence methodology was used for mRNA quantification. TaqMan probe and primers were purchased as TaqMan gene expression assays-on-demand from Thermofisher. Target sequences for each primer are shown in Appendix A. Each gene was tested in triplicate and normalized to the expression of the housekeeping transcript GAPDH.

### 2.5. Bone Histomorphometry

To conduct histomorphometric measurements, the proximal part of the tibia was fixed in 70% ethanol overnight at 4 °C, dehydrated and embedded in 100% methylmethacrylate (Merck, Hohenbrunn, Germany). Tibia sections (5 µM thick) were Goldner stained to determine static bone parameters, including total bone area, mineralized bone area, osteoid area/-width, osteoid and eroded perimeter and osteoblast perimeter and osteoclast perimeter, using the Axiovision image analysis software (Release 4.5, Carl Zeiss, Oberkochen, Germany). Unstained tibia sections (10 µM thick) were used to calculate the distance between and length of double tetracycline and demeclocycline labels by fluorescence microscopy.

Out of these primary data, bone formation rate and mineralization lag time was calculated according to Dempster et al. [26]. See Appendix A for explanation and calculations of static and dynamic bone parameters.

### 2.6. Statistical Analysis

Statistical comparisons were made by non-parametric testing (Prism 8.1.1, GraphPad Software Inc., San Diego, CA, USA). To investigate the statistical difference between groups at one time-point, a Kruskal–Wallis test multiple comparisons was applied. The *p*-value was adjusted by Bonferroni correction. Representative data are presented as mean ± SEM (standard error of mean) and considered significant when the adjusted *p*-value ≤ 0.05. In the Appendix A show the exact *p*-value of parameters with statistical significance. A power analysis was used to determine sample size. With an expected standard deviation of 16 mg/g tissue, 12 animals per group are required to obtain a power of 90%. One vehicle-treated CKD rat died before sacrifice and was excluded from the study.

## 3. Results

### 3.1. Adenine-Fed Rats Developed a Stable Chronic Renal Failure Based on Serum Creatinine, Phosphorus and Calcium Levels

The groups who received a 0.75% adenine diet developed a stable chronic renal failure, as indicated by a significant increase in serum creatinine and phosphorus levels as well as a significant decrease in serum calcium levels as compared to rats on a control diet (Table 1). Based on urine and serum creatinine levels, creatinine clearance was calculated and showed a significant decrease in CKD rats receiving the 0.75% adenine diet versus rats on a control diet (Table 1). No differences in this respect were noted between the CKD groups. The phosphaturic hormone FGF23 was significantly increased in CKD rats treated with SBI-425 and SBI-425 + PPi as compared to control rats. Furthermore, no differences between the groups for serum alkaline phosphatase activity and plasma PPi levels were observed. The alkaline phosphatase activity was also measured in the aortic tissue showing no significant changes between groups (control: 9.40 ± 1.75 U/g wet tissue; vehicle: 12.68 ± 2.09 U/g wet tissue; TNAPi: 13.69 ± 2.43 U/g wet tissue; PPi: 14.08 ± 1.52 U/g wet tissue; TNAPi + PPi: 18.47 ± 1.66 U/g wet tissue). As alkaline phosphatase plays an important role in liver function, serum aspartate aminotransferase (AST) and alanine aminotransferase (ALT) levels were measured as indicators for liver toxicity. Serum AST and ALT levels did not differ significantly between all groups.

### 3.2. Exposure to PPi, but Not TNAP Inhibition, Prevented the Development of Arterial Media Calcification

Adenine-induced chronic renal failure triggered the development of arterial media calcification since a significant higher calcium content in the aorta, carotid and femoral artery was observed in vehicle-treated CKD rats versus control rats. In addition, whilst CKD rats exposed to SBI-425 developed arterial calcification, CKD rats treated with PPi with or without SBI-425 showed a significant decrease in the calcium content in the aorta, carotid and femoral artery (Figure 1). Arterial calcium content of arteries from vehicle- versus SBI-425-treated animals was not statistically different. Arterial calcium content measurements were corroborated by von Kossa staining of the aortic tissue of the different treatment groups (Figure 2) since tissue sections of vehicle- and SBI-425-treated animals showed the most prominent von Kossa staining, without statistical difference between those groups (see Appendix A).

### 3.3. mRNA Expression of Genes Involved in CKD-Related Arterial Media Calcification

During arterial media calcification VSMCs transdifferentiate into bone-like cells; therefore, the mRNA expression of genes involved in this process was evaluated (Figure 3). The mRNA expression of *Sm22-alpha*, a specific VSMC marker, was decreased in all CKD groups as compared to control rats. The aortic mRNA expression of osteo-/chondrogenic marker genes *Sox9* and *Runx-2* showed an increasing trend in all CKD groups with exception of PPi-treated CKD rats and became significant for SBI-425-treated CKD rats. Except for the groups treated with PPi, the aortic mRNA expression of type III sodium-dependent phosphate transporter 1 (*Pit-1*) significantly increased in the presence of CKD, while expression of *Pit-2* in the presence of CKD significantly decreased. Additionally, the mRNA expression of genes involved in the homeostasis of Pi/PPi balance, including *Tnap*, *Npp1*, *Npp3* and *Phospho1* were analyzed (Figure 4). Aortic mRNA expression of *Tnap* decreased in CKD groups as compared to the control group. In general, this was also the case for mRNA levels of alternative phosphate generating enzymes, *Ntpd1* and *Phospho1*, only for PPi-treated rats *Phospho1* mRNA stayed at control levels. Aortic expression of *Npp1* and *Npp3* on the other hand was increased as a result of CKD, which was, however, not the case for *Npp1* expression in PPi-treated rats.

### 3.4. Treatment with SBI-425 with or without PPi Administration Worsens Bone Metabolism in a CKD Context in the Rat

Since arterial media calcification resembles physiological bone mineralization, it is imperative to investigate whether PPi, TNAP inhibition and the combination of both treatments affected bone metabolism. Hereto static bone parameters were investigated in all groups (Figure 5 and Figure 6). Inherent to CKD, bone and mineralized area decreased in all CKD groups (not significantly in PPi-treated groups) versus rats with normal renal function. Furthermore, a trend towards increased osteoid area, –width and –perimeter was found in SBI-425-treated CKD rats, which became significant in CKD rats treated with a combination of TNAP inhibitor and PPi as compared to both control rats and vehicle-treated CKD rats. In addition, CKD rats treated with SBI-425 with or without PPi showed a significant increase in the osteoblast perimeter versus control rats. The osteoclast perimeter significantly decreased in CKD rats treated with SBI-425 + PPi versus CKD rats treated with vehicle. Bone formation rate was not significantly augmented in rats with CKD as compared to control rats. Mineralization lag time was significantly increased in CKD rats treated with SBI-425 + PPi versus CKD rats treated with vehicle.

## 4. Discussion

Arterial media calcification is a master of camouflage by disguising itself as physiological bone mineralization, which makes it extremely difficult to find therapies that target it efficiently and, most of all, safely, i.e., without affecting bone metabolism. Certainly CKD patients, an important target population for arterial media calcification, suffer from bone mineralization defects [24]. Our study investigated whether the TNAP inhibitor SBI-425, PPi and the combination of SBI-425 + PPi was able to prevent the development of arterial media calcification in a rat model of CKD without aggravating bone metabolism.

The animals received an adenine enriched diet to induce chronic renal failure, which was successfully accomplished as indicated by the presence of hyperphosphatemia, hypocalcemia and increased serum creatinine levels. The degree of CKD in those animals was similar to CKD patients in stage 4–5 [27,28]. Our CKD rat model gives a suitable reflection of the presence of arterial calcification as these CKD patients develop calcifications in the medial layer of the arterial wall with a prevalence of 40–70% [29,30,31], and is thus comparable to our vehicle-treated CKD rats with calcification in 58% (7/12) of the animals. The degree of calcification, expressed as the amount of calcium/g tissue, in the aorta and peripheral arteries was 10–20 times control values. Furthermore, calcification in the aorta and peripheral arteries was significantly inhibited by PPi and SBI-425 + PPi. Interestingly, recent studies showed that orally administered PPi was able to prevent connective tissue calcification in mouse models of pseudoxanthoma elasticum and general arterial calcification of infancy [32,33]. SBI-425 alone, however, did not prevent arterial calcification in the rat CKD context of the current experiment, in contrast to our observations in the rat model of warfarin-induced arterial calcifications where an identical SBI-425 dose was used [22]. This could possibly be explained by the fact that these animals have an intact renal function and arterial calcification in the warfarin model is much less severe.

In a next step, aortic mRNA expression profile of VSMC and bone-like marker genes revealed an up- and downregulation of *Sox9/Runx2* (osteo-/chondrogenic marker genes) and *Sm22-alpha* (VSMC marker gene), respectively, in vehicle and TNAP inhibitor-treated CKD rats versus control rats, favoring a VSMC phenotypic switch into bone-like cells. The fact that *Sox9/Runx2* expression seemed to be less increased in PPi-treated animals is in agreement with their lower calcification grade.

PIT-1, a type III sodium-dependent phosphate transporter is involved in osteo-/chondrogenic transition of VSMCs [34] as well as in endoplasmic reticulum stress [35] and apoptotic events [36], all processes associated with the induction of arterial media calcifications [9]. Administration of PPi prevented the CKD-induced upregulation of aortic *Pit-1* mRNA, suggesting that PPi, by altering the concentration of phosphate transporter inhibitors (such as, e.g., uremic toxins) [37], either directly or indirectly inhibits expression of this important stimulator of arterial calcification. Further research is necessary to unravel the exact relationship between PPi, PIT-1 and the development of arterial media calcification. Next, we investigated *Pit-2* expression, another sodium-dependent phosphate transporter that has been shown to be expressed in VSMCs [38]. Redundancy by *Pit-2* could maintain arterial calcification development in mice with targeted deletion of *Pit-1* in VSMCs [39]. On the other hand, data from an experimental mouse study revealed that *Pit-2* haplo-insufficiency enhanced the development of arterial calcification in CKD [40]. Interestingly, in our study, aortic *Pit-2* mRNA expression was significantly lower in all CKD groups versus control rats, thereby implying that the presence of CKD alters *Pit-2* gene expression, which, in turn, might influence the arterial calcification process. Hence, the exact role of PIT-2 in CKD-related arterial calcification needs to be further investigated. Subsequently, the mRNA expression profile of other genes involved in the PPi/Pi homeostasis were further investigated. The aortic tissue of normal rats expresses three important ecto-nucleotidases including *Npp1, Npp3* and *Tnap* [41]. In the present study, aortic mRNA expression of *Npp1* was increased as a result of CKD, which was not the case for PPi-treated CKD rats (both PPi alone and combined with SBI-425 had significantly lower values as compared to SBI-425 alone). These results suggest that (i) *Npp1* mRNA expression is higher in the presence of arterial calcification as observed in the vehicle and SBI-425-treated CKD groups and (ii) increasing levels of PPi induce a negative feedback on the NPP1 enzyme expression. The latter is in line with previous studies that have shown that PPi is able to down-regulate *Npp1* expression [42,43]. Additionally, NPP3 has a more complex role in the process of mineralization since NPP3 has been implicated in both the production as well as the hydrolysis of PPi [41]. Interestingly, in the present rat study, chronic renal failure induced an increased aortic *Npp3* expression (significant in the SBI-425-treated animals), which was not observed for the aortic mRNA expression of the TNAP enzyme. Consequently, these data suggest that in the present study and, thus, in the context of a CKD rat model, elevated *Npp3* and *Npp1* expression, rather than *Tnap*, might be responsible for the hydrolysis of PPi into Pi in the aortic tissue and, by these means, the stimulation of the arterial calcification process. A study of Villa-Bellosta et al. showed that during the early stages of arterial calcification, the mRNA expression of *Tnap* and *Npp1*, respectively, is also decreased and increased, while, in the later stages of arterial calcification, *Tnap* mRNA expression becomes upregulated [44]. However, while little data can be found on the role of *Npp3* in arterial calcification at present, *Npp1* has been shown to behave as a potent phosphatase in the absence of TNAP [45]. Furthermore, PHOSPHO1 is also implicated in controlling the aortic PPi/Pi ratio through cleavage of phospholipids, originating from calcified matrix vesicles, with the release of Pi [46,47]. The *Phospho1* mRNA expression seemed to be lower as a result of CKD induction, which was also the case for the mRNA expression of *Ntpd1*, an enzyme that converts ATP into ADP and releases Pi. Taken together, a similar expression profile in CKD rats of the Pi generating enzymes TNAP, PHOSPHO1 and NTPD1 was observed, suggesting that CKD-related factors including high levels of phosphate and calcium [48] could inhibit these Pi generating enzymes.

In line with other studies using different animal models to target the PPi/Pi ratio by either TNAP inhibition or PPi supplementation, we found no differences in circulating PPi levels between the different study groups [19,32,49,50]. It is worth noting that the PPi plasma values reflect the whole-body PPi metabolism. PPi is released during multiple biochemical reactions (i.e DNA synthesis, nucleotide hydrolysis, amino acid synthesis, glycogen and urea synthesis, cyclic GMP synthesis, etc.). Moreover, the study of Dedinskzi [32] clearly showed that administration of exogenous PPi is rapidly cleared from the plasma as, after 2 h of administering a 40 mg/kg or 150 µmol/kg PPi dose, the plasma PPi concentration had almost returned to baseline values. Nevertheless, based on the aortic mRNA expression profiles of *Npp1, Ntpd1* and *Tnap*, an increase in aortic PPi levels of vehicle- and TNAP inhibitor-treated CKD rats may reasonably be expected given the decrease in Pi-generating enzymes TNAP and NTPD1 and increase in PPi-generating enzymes *Npp1* and *Npp3*. However, as mentioned above, it has been proposed that both *Npp1* and *Npp3* can break down PPi into Pi and, thus, can also act as Pi-generating enzymes. Unfortunately, measuring arterial PPi, in our hands, as well as others [19,32,49,50], turns out to be technically impossible, due to rapid PPi hydrolysis during tissue handling/digestion. However, the importance of the local effect of TNAP on PPi aortic levels has been demonstrated in patients with hypophosphatasia. These patients have a loss-of-function mutation in the gene encoding TNAP leading to impaired bone mineralization, histologically expressed as osteomalacia. Only administration of a mineral-targeted recombinant TNAP, rather than systemic TNAP, improves the disease, most probably due to a local correction of the PPi concentrations at the site of calcification [51].

Next to its role in regulating the PPi/Pi homeostasis, TNAP is also expressed in the liver to mediate anti-inflammatory actions including dephosphorylation of the bacterial endotoxin lipopolysaccharide [52]. Biochemical analysis of the serum showed that no differences were found for ALT and AST enzymatic activity (two markers of liver toxicity), in all CKD groups versus control rats, demonstrating that treatment with PPi and the combination of SBI-425 + PPi inhibited CKD-induced arterial media calcification without affecting liver function. In contrast to the weak, non-selective inhibitors (i.e., levamisole and theophylline) against alkaline phosphatase isozymes (i.e., placental-, germ cell-, intestinal- alkaline phosphatase and TNAP), Pinkerton et al. showed that SBI-425 is a potent, selective and strong binding inhibitor against solely the TNAP isozyme [16].

During chronic renal failure, patients develop bone mineralization abnormalities due to changes in hormone expression (i.e., PTH, FGF23 and vitamin D) that control mineral homeostasis and osteoblast/-clast function [23]. Static bone parameters were analyzed to evaluate bone homeostasis. The presence of renal insufficiency resulted in a significant decrease in bone- and mineralized area without significant changes in osteoid area/-width between CKD animals treated with vehicle versus control animals. Moreover, the bone formation rate was increased in rats with renal impairment. Although it was not significant, several CKD rats had higher bone formation rates than the upper detection limit. Therefore, these values were shown in Figure 5D as “high value”. Treatment with SBI-425, especially in combination with PPi, significantly increased osteoid area/-width and osteoblast perimeter while mineralized area remained unaltered as compared to vehicle-treated CKD rats. Furthermore, the combination therapy SBI-425 with PPi induced a significantly higher mineralization lag time or the time interval between the deposition of osteoid and its subsequent mineralization. These data suggest that treatment with SBI-425 in combination with PPi induces inappropriate mineralization of the osteoid, leading to the histological picture of the so-called ‘mixed renal osteodystrophy’ featuring the histological characteristics of both an increased bone turnover and osteomalacia, which probably points to a dysfunction of osteoblasts resulting from an insufficient inorganic phosphate supply mediated by SBI-425-induced TNAP inhibition. Interestingly, daily treatment with PPi alone did not worsen CKD-related bone formation/-mineralization defects in our study as one may reasonably assume that in the absence of a TNAP inhibitor, the enzymatic TNAP-mediated PPi breakdown remains intact in the bone, through which the PPi/Pi ratio is kept stable. Taken together, daily treatment with SBI-425 with or without PPi worsens CKD bone remodeling defects in the rat characterized by disorganized bone mineralization, which, in a clinical setting, may ultimately result in an increased fracture risk. Furthermore, compared to control animals, serum FGF23 was only increased in rats treated with SBI-425 with or without PPi, suggesting that both therapies mediate inappropriate inorganic phosphate built-up to the bone, raising the phosphaturic hormone FGF23 levels in order to restore phosphate homeostasis.

The fact that SBI-425 did not inhibit the arterial calcifications in our rat CKD model may be due to (i) the fact that unlike in the mouse CKD model, where mRNA expression of *Tnap* was upregulated ~3-fold [20], this was not seen in our rat CKD model, (ii) the use of an overly low concentration of the compound, which reasonably cannot be increased, given the effects observed at the level of the bone, and (iii) the earlier observations that plasma alkaline phosphatase activity in humans and mice can be significantly inhibited by SBI-425, whilst this is much less the case in rats [53] (see also Appendix A). This is in agreement with the total plasma alkaline phosphatase activity that was unaltered in the groups receiving SBI-425 treatment. The same observation was done, however, in our study investigating the effect of SBI-425 in a warfarin rat model [22], in which a clear inhibition of the arterial calcifications was seen, questioning the role/importance of systemic TNAP activity in the process of arterial calcification.

Independently of the effects on arterial calcification, however, in this study, we found that SBI-425 treatment, either in combination with PPi or not, triggers bone mineralization defects. Clearly, potential side effects on bone mineralization will need to be assessed in any clinical trial aimed at modifying the Pi/PPi ratio in CKD patients who already suffer from a compromised bone status.

## 5. Patents

Pinkerton A.B. and Millan J.L. are co-inventors on a patent application covering SBI-425 (PCT WO 2013126608).

## Figures and Tables

**Figure 1 pharmaceutics-13-01138-f001:**
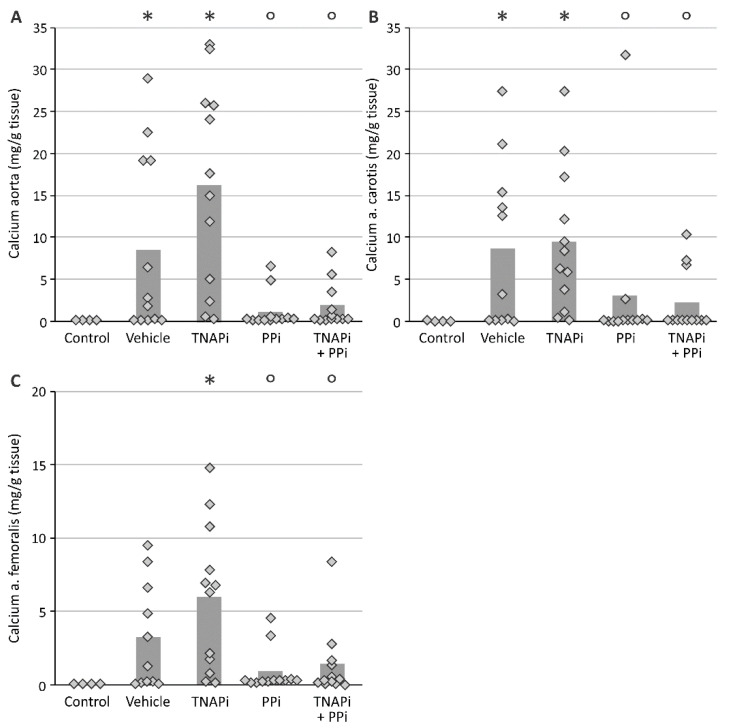
Evaluation of the presence of arterial calcification. Calcium content of (**A**) aorta, (**B**) carotid artery, (**C**) femoral artery. Histogram and dots represent mean and individual values, respectively. * *p* < 0.05 versus control, ° *p* < 0.05 versus TNAPi.

**Figure 2 pharmaceutics-13-01138-f002:**
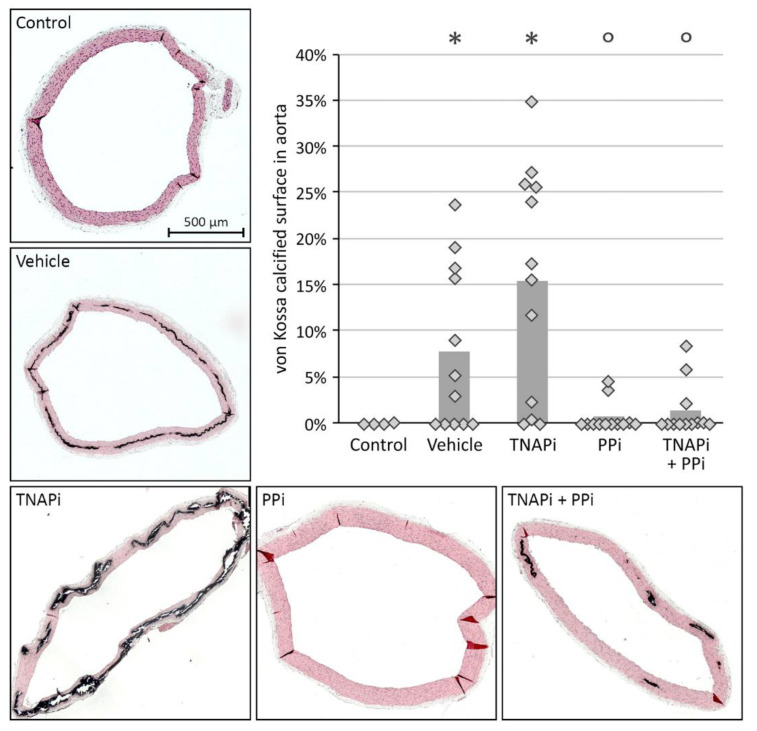
Visualization of the aortic calcifications. Representative Von Kossa stained aortic sections of control rat and vehicle, TNAP inhibitor, PPi and TNAP inhibitor + PPi-treated CKD rats. % calcified aortic area is shown in the graph with histogram and dots representing mean and individual values, respectively. * *p* < 0.05 versus control, ° *p* < 0.05 versus TNAPi.

**Figure 3 pharmaceutics-13-01138-f003:**
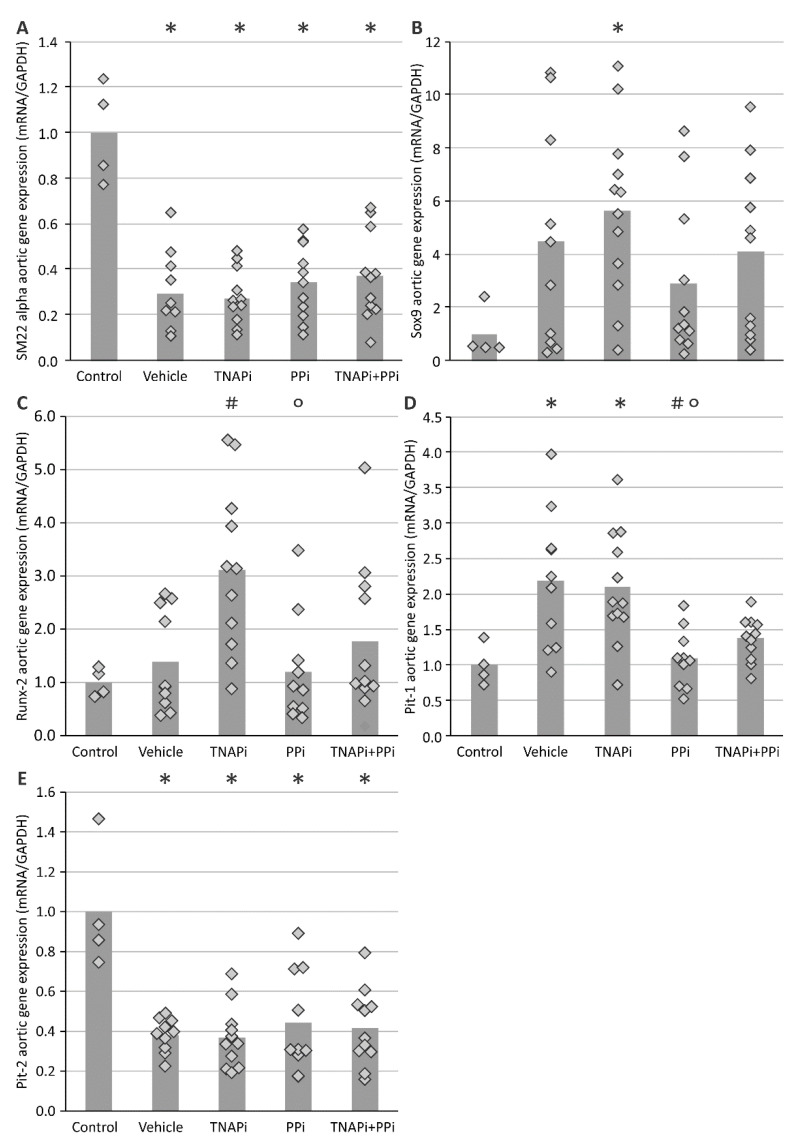
mRNA aortic gene expression of osteo-/chondrogenic marker genes; (**A**) SM22 alpha, (**B**) Sox9, (**C**) Runx-2, (**D**) Pit-1 and (**E**) Pit-2. Histogram and dots represent mean and individual values, respectively. * *p* < 0.05 versus control, # *p* < 0.05 versus vehicle, ° *p* < 0.05 versus TNAPi.

**Figure 4 pharmaceutics-13-01138-f004:**
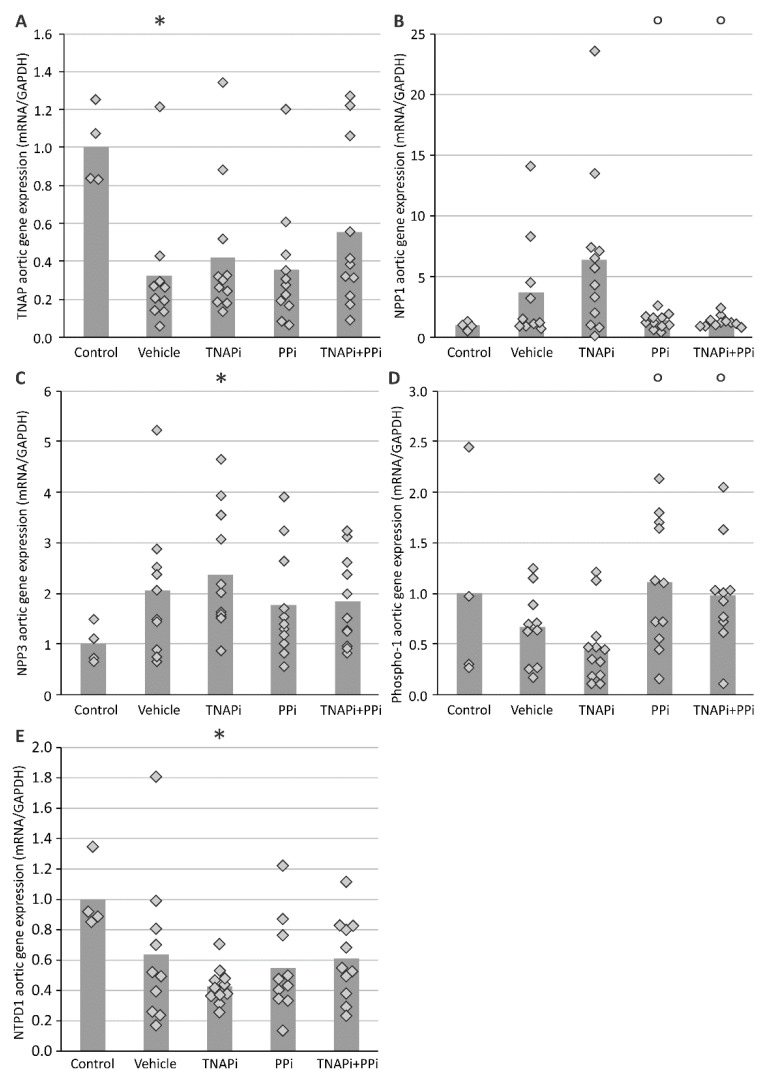
mRNA aortic gene expression of genes involved in PPi/Pi homeostasis; (**A**) TNAP, (**B**) NPP1, (**C**) NPP3, (**D**) Phospho-1 and (**E**) NTPD1. Histogram and dots represent mean and individual values, respectively. * *p* < 0.05 versus control, ° *p* < 0.05 versus TNAPi.

**Figure 5 pharmaceutics-13-01138-f005:**
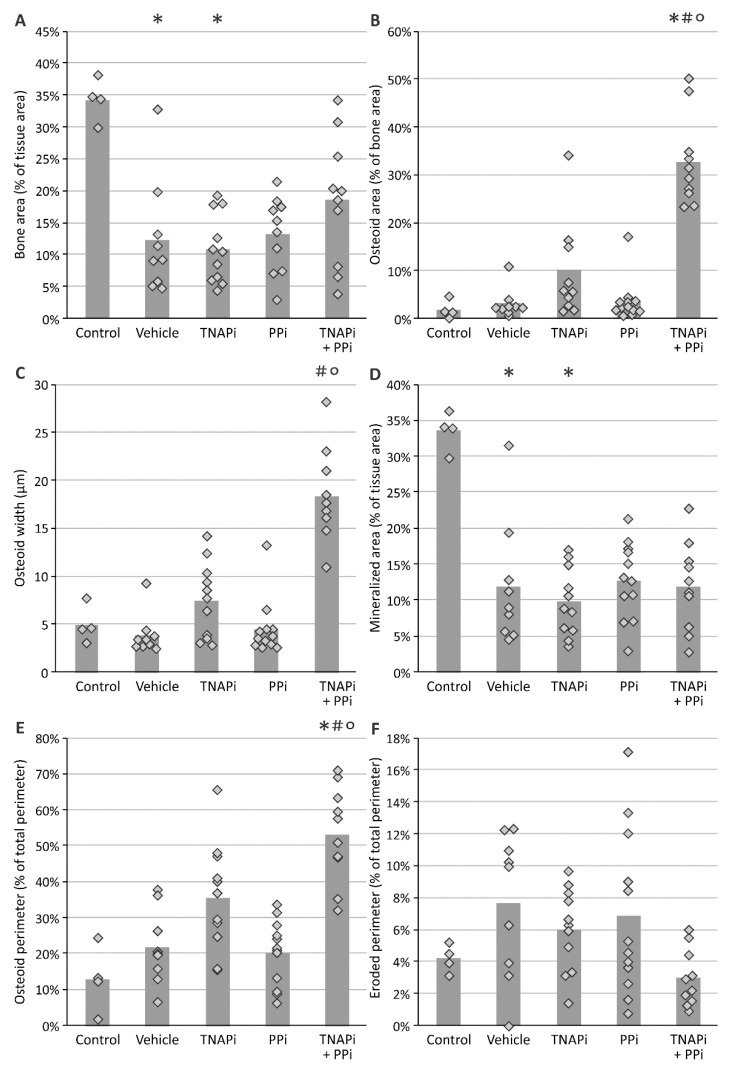
Evaluation of static bone parameters. (**A**) bone area, (**B**) osteoid area, (**C**) osteoid width, (**D**) mineralized area, (**E**) osteoid perimeter and (**F**) eroded perimeter. Histogram and dots represent mean and individual values, respectively. * *p* < 0.05 versus control, # *p* < 0.05 versus vehicle, ° *p* < 0.05 versus PPi.

**Figure 6 pharmaceutics-13-01138-f006:**
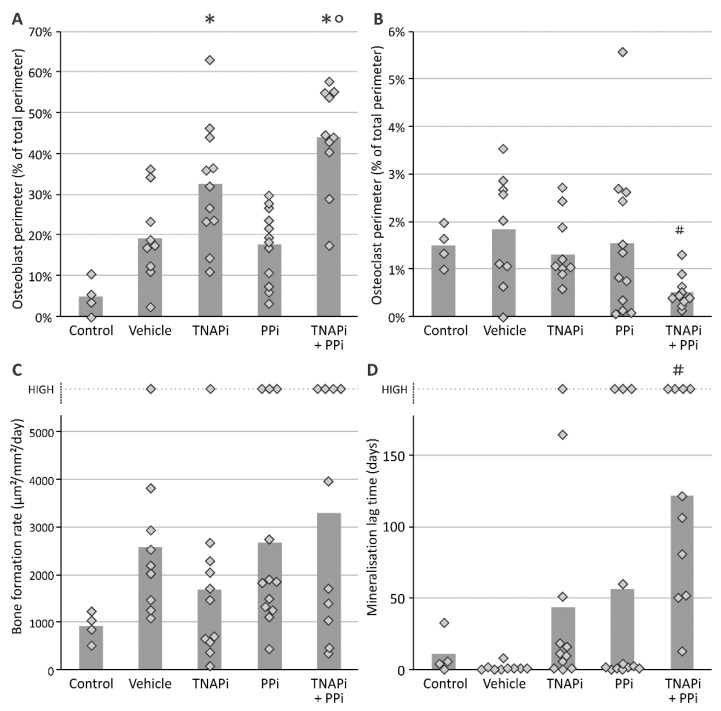
Evaluation of static bone parameters and bone formation rate. (**A**) osteoblast perimeter, (**B**) osteoclast perimeter, (**C**) bone formation rate and (**D**) mineralization lag time with dark dots indicating values above detection limit. Histogram and dots represent mean and individual values, respectively. * *p* < 0.05 versus control, # *p* < 0.05 versus vehicle, ° *p* < 0.05 versus TNAPi.

**Table 1 pharmaceutics-13-01138-t001:** Serum/plasma biochemical parameters.

Parameter	Control	Vehicle	TNAPi	PPi	TNAPi + PPi
Number of rats	4	11	12	12	12
Creatinine (mg/dl)	0.70 ± 0.09	6.38 ± 0.43 ^a^	6.27 ± 0.54 ^a^	6.28 ± 0.37 ^a^	6.40 ± 0.36 ^a^
Creatinine clearance (mL/min)	2.42 ± 0.45	0.15 ± 0.02 ^a^	0.11 ± 0.01 ^a^	0.14 ± 0.01 ^a^	0.12 ± 0.01 ^a^
Phosphorus (mg/dl)	8.85 ± 1.64	23.36 ± 0.80 ^a^	22.86 ± 0.69 ^a^	24.72 ± 1.28 ^a^	23.87 ± 1.32 ^a^
Calcium (mg/dl)	13.74 ± 0.34	11.29 ± 0.25 ^a^	11.63 ± 0.26	11.28 ± 0.24 ^a^	11.41 ± 0.54 ^a^
FGF23 (pg/µL)	0.22 ± 0.02	83 ± 26	130 ± 22 ^a^	55 ± 11	116 ± 23 ^a^
ALP (U/L)	182 ± 12	127 ± 9	170 ± 20	122 ± 12	159 ± 11
PP_i_ (µM)	6.2 ± 1.56	8.2 ± 1.30	7.5 ± 1.39	6.5 ± 1.06	10.4 ± 1.20
AST (U/L)	78.5 ± 2.2	72.6 ± 5.6	76.0 ± 6.9	79.2 ± 6.4	82.2 ± 5.8
ALT (U/L)	37.5 ± 2.8	18.6 ± 2.1	17.8 ± 2.5 ^a^	23.5 ± 5.5	19.0 ± 2.1

Mean ± SEM, ^a^: *p* < 0.05 versus control. FGF23: fibroblast growth factor 23; ALP: alkaline phosphatase; AST: aspartate aminotransferase; ALT: alanine aminotransferase.

## Data Availability

Data that support findings of this study are available upon request from the authors.

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
