# Peer review of "Chronic Kidney Disease-Induced Arterial Media Calcification in Rats Prevented by Tissue Non-Specific Alkaline Phosphatase Substrate Supplementation Rather Than Inhibition of the Enzyme"

_pharmaceutics, 2021, doi:10.3390/pharmaceutics13081138_

Round 1
Reviewer 1 Report
The Article manuscript from Opdebeeck et al. “Chronic kidney disease induced arterial media calcification in rats prevented by tissue non-specific alkaline phosphatase substrate supplementation rather than inhibition of the enzyme” describes new findings on the function of a rather novel TNAP inhibitor SBI-425 in rat and its influence on arterial calcification. The authors present their results in a logical order, describe the experimental methods in good detail, and discuss their findings in accordance with up-to-date knowledge. The authors present their novel findings nicely and in a well-structured way, so that only a few minor points of criticism have to be raised:
Major points:
- line 190 and 229: Supplementary data table S3 and data table S2-S4 are missing from the current manuscript. Might be a pdf production problem or artifacts originating form an updated manuscript version.
Minor points:
- line 29-32, 38-41, 41-45, 75-78: Statements might need additional reference(s) or does the references at the end of the section include all previous sentences? Please recheck and add references if needed.
- line 83, ff: The animal experiments are described in decent detail, but please state if animal experiments have been performed according to ARRIVE guidelines. You might want to add (as requested by the journal) the ARRIVE checklist to the supplement (https://arriveguidelines.org/resources/questionnaire.).
-line 93-97: A simple scheme for further clarification of the different treatments would be highly appreciated (supplement figure).
-line 93: “10mg/kg/dag” should be day
-line 138: Relocate “(Kainos Laboratories, Tokyo, Japan)” to the end of the sentence, as it should be referring to the ELISA and not to FGF23 as its current position implies.
- line 156, ff: Please add target sequences of used (TaqMan) primers to the supplement.
- Table 1 and fig. 1 to 6: Given group names in table 1 and in all figures differ and make comparison of data less intuitive. Please harmonize group names, e.g. by stating or renaming group names in table 1.
- Different lines in the manuscript, e.g. line 157-164: Please correct rat gene names throughout the document. “Gene symbols generally are italicised, with only the first letter in uppercase and the remaining letters in lowercase (Shh). Italics are not required on web pages. Protein designations are the same as the gene symbol, but are not italicised and all are upper case (SHH)” http://www.informatics.jax.org/mgihome/nomen/gene.shtml
- Fig. 1 to 6: Statistic marker (*, #, o) are rather small and hard to see, as they are merged with the top line. Optical enhancement by a larger font size or moving symbols above the upper line could be helpful.
- line 141: Hereto a Von Kossa’s method: „a“ can be deleted as it is superfluous
- line 178: The “osteoid and eroded perimeter and osteoblast perimeter and osteoclast perimeter“ might need a bit more explanation within the methods section. To me it is unclear what is measured by these values. Maybe a supplement figure showing these parameters would be helpful.
- line 179: Ten µm = 10 µm
- line 206 to 208: U/g should be written in a similar way.
- line 213: a p<0.05 versus control should be a: p<0.05 versus control instead (: is missing after the a)
- line 310: “patients in stage []” missing stage number ?
- line 333: cations[3]. missing space
- line 365: the mRNA expression of TNAP and NPP1 respectively are also decreased and increased: it should be is decreased and increase as the verb refers to mRNA expression which is singular
- line 373: “anenzyme” space missing
- line 379: different study groups[12,25,42,43] space missing between groups and [
- line 390: „…and thus can act as Pi-generating enzymes also.“ In my opinion you should replace “also“ with “..,too“ or rearrange the sentence “can also act as…“
- line 406 ff, discussion: A short comment/comparison would be appreciated on how SBI-425 function is different to other commercially available TNAP inhibitors (like Levamisole or the sterically very similar CAS496014-13-2 aka. MLS-0038949).
- line 441: “much less the case in rats[46]” double and missing space in pdf.
Reviewer 2 Report
- In the paper “Chronic kidney disease induced-arterial media calcification in 2 rats prevented by tissue non-specific alkaline phosphatase sub-3 strate supplementation rather than inhibition of the enzyme” by Opdebeeck et al., the authors aimed to investigate whether a TNAP inhibitor, Ppi or their combination might inhibit arterial media calcification in a uremic animal model. The paper is very well-written with solid design, structured methodology and interesting results. My only concern is that the parts in introduction and discussion regarding calcification and calcification inhibitors in CKD are limited and with outdated references. I would suggest that the authors would enrich these parts with more details and cote dated papers (recommended papers are doi 10.3390/jcm9082359, 10.1007/s00109-021-02037-7, 10.1016/bs.acc.2020.02.004)
